# Optimal Configuration of Integrated Energy System Based on Energy-Conversion Interface

**Zicong Yu** [iD]**, Xiaohua Yang, Lu Zhang, Yongqiang Zhu \*, Ruihua Xia and Na Zhao**

School of Electrical and Electronic Engineering, North China Electric Power University, Beijing 102206, China;
yuzicong@ncepu.edu.cn (Z.Y.); xiaohuayang@ncepu.edu.cn (X.Y.); lucy_ncepu@163.com (L.Z.);
supplyports@sina.com (R.X.); zhaona0726@163.com (N.Z.)
\* Correspondence: zyq@ncepu.edu.cn; Tel.: +86-139-1000-2860

**Abstract:** Aiming at the optimal configuration of a regional integrated energy system (IES), this paper proposes an energy-conversion interface (ECI) model that simplifies the complex multienergy network into a multi-input–multioutput dual-port network, consequently achieving the energy-coupling relationship between the energy-supply side and the demand side. An optimized configuration model of the ECI was constructed by considering economic performance, such as device-installation cost, operation and maintenance cost, and environmental cost, as well as energy-saving performance, such as energy-utilization efficiency. Then, the ECI optimal-configuration model was established by taking a campus in northern China as an example. To verify the validity of the model, device planning quantity and daily energy scheduling of the integrated energy system of the campus were obtained by solving the model with the particle-swarm optimization method. Finally, sensitivity analysis of the system to energy prices and the reweight approach for the targets are also given in this paper, providing a decision-making basis for system planning.

**Keywords:** integrated energy system; energy-conversion interface; optimal configuration

## 1. Introduction

Lack of co-ordination and co-operation between traditional energy systems, such as power systems, natural-gas systems, and thermal systems, results in dissatisfying advantageous complementary and synergy energy benefits, because these energy systems are separately planned and operated. An Integrated Energy System (IES) has different energy forms collaborating in development, conversion, storage, transportation to meet the needs of terminal consumers.

To improve energy efficiency and economic benefits, reduce environmental influence, and satisfy various energy demands at the same time, an IES utilizes advanced technologies for load forecasting, data processing, and innovative management in a certain area to co-ordinate, optimize, and intelligently manage multiple energy sources, such as electricity, natural gas, petroleum, and coal [1–3].

At present, the optimization problems of regional integrated energy systems can be divided into two categories: (1) optimizing the capacity and model of devices for a given regional IES and (2) optimizing energy flows for existing system topologies. For the former, Bahrami et al. [4] co-ordinated the capacity of cogeneration units, absorption chillers, gas boilers, and other devices. Singh et al. [5] used the genetic algorithm to optimize the location and capacity of IES power supply with the aim of minimizing system network loss. Xu et al. [6] planned devices in building an integrated energy system and analyzed the economic benefits of combined heat and power (CHP). For the latter, Li et al. [7] proposed a mixed-integer nonlinear model to optimize the system with the lowest operating cost. Ress et al. [8] estimated the thermal and electrical flows within the system by analyzing the thermoelectric-energy system model on the distribution side. Rezvan et al. [9] optimized

the device output of the system after dealing with load volatility by a Monte Carlo simulation. Yang et al. [10] proposed an optimal dispatching model of a electrothermal multienergy system with power-to-gas (P2G) facilities on the basis of considering the characteristics of P2G facilities, power systems, natural-gas systems, and heating systems. Thieblemont et al. [11] put forward sensible thermal-energy storage (TES) systems that can shift a portion or all of it to off-peak periods to help reduce peak demand and electrical-grid stress.

With the development of IES, an increasing number of scholars have begun to turn their focus to a comprehensive benefit evaluation of the system. Ruan et al. [12] considered the economic cost-saving rate, primary energy-saving rate, pollution-gas emission-reduction rate, and other indicators, analyzed the four types of buildings, and finally concluded that buildings with higher heat-to-electric ratio were more suitable for energy supply by way of combined cooling, heating, and power. Wang et al. [13] established a comprehensive IES indicator system that included economic, environmental, and social benefits, and combined the entropy weight method and the analytic hierarchy process to give weights of the indicators. Bai et al. [14] proposed a comprehensive energy-utilization assessment indicator for multienergy flow characteristics existing in IES to reflect the system's acceptance level and energy-efficiency level of renewable energy.

In summary, there are two difficulties in the study of IES: (1) simultaneously planning the topology and IES device capacity or model, that is, IES planning starts from scratch; and (2) combining IES energy-dispatch optimization with planning devices to support each other.

In response to the above problems, this paper introduces the concept of the Energy Conversion Interface (ECI) [15–17], which is applied to regional load-side IES, as shown in Figure 1. The ECI connects to a multienergy transmission network and the demand side, which enables the distribution and conversion of multiple energy sources and regulates energy flowing into the demand side.

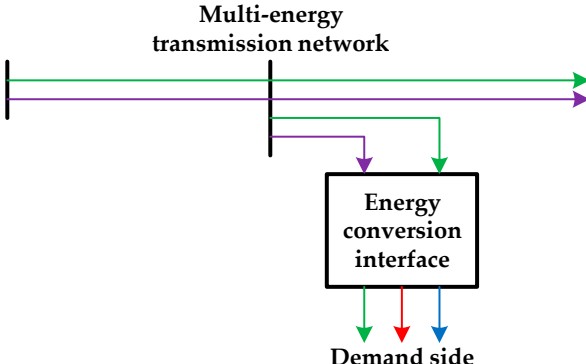

**Figure 1.** Integrated energy system schematic.

First, this paper establishes an energy input–output coupling model by deducing the energy input–output relationship of the energy-conversion interface, and classifying and refining the energy-conversion devices. The model has good ductility and clarifies the relationship between system energy flow and devices. Second, on the basis of economic indicators such as device-installation cost, operation and maintenance cost, and annual cost, as well as energy-saving indicators, such as energy-utilization efficiency, an optimized ECI configuration model was established. The model has the following innovations: (1) IES planning starts from scratch and (2) energy-optimization dispatch and device quantity are jointly planned.

Finally, an input–output coupling model was constructed for a campus in northern China, and the model was solved with the particle-swarm algorithm method. During this procedure, the planned quantity of various energy-conversion devices and energy-storage devices, as well as the daily energy schedule, were obtained. System-sensitivity analysis to energy prices and the reweight approach for targets are also given in this paper, providing a decision-making basis for system planning.

## 2. Energy-Conversion Interface Model

The energy-conversion interface is shown in Figure 2, where $E_m$ is the $m$-th type of input energy, $L_n$ is the $n$-th type of output energy, and $C$ is the transformation matrix. The mathematical relationship between the supply side and demand side is deduced and analyzed in this section.

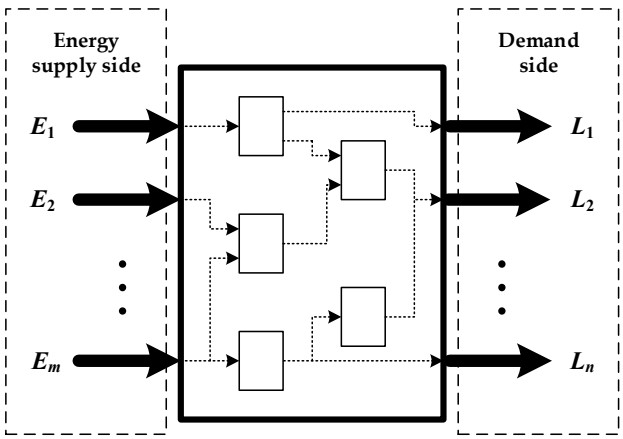

**Figure 2.** Integrated energy system schematic.

### 2.1. Simple Input–Output Coupling Model

A simple input–output coupling model can be constructed when all the devices' output energy inside the ECI goes to the demand side. The energy-conversion process inside the ECI is shown in Figure 3.

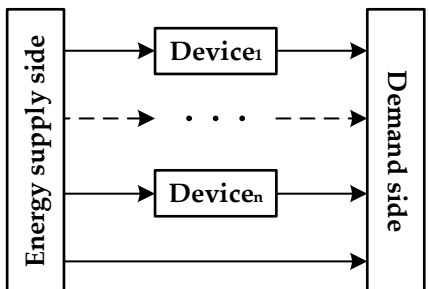

**Figure 3.** Simple energy-conversion interface (ECI) energy-flow diagram.

In a simple ECI, the relationship between Input **E** and Output **L** can be expressed by Equation (1). For a system with $m$ types of input energy and $n$ types of output energy, the input–output relationship can be specifically expressed by Equation (2).

$$\mathbf{L} = \mathbf{C} \cdot \mathbf{E} \tag{1}$$

$$\begin{bmatrix} l_1 \\ l_2 \\ \vdots \\ l_n \end{bmatrix} = \begin{bmatrix} c_{11} & c_{12} & \cdots & c_{1m} \\ c_{21} & c_{22} & \cdots & c_{2m} \\ \vdots & \vdots & \ddots & \vdots \\ c_{n1} & c_{n2} & \cdots & c_{nm} \end{bmatrix} \begin{bmatrix} e_1 \\ e_2 \\ \vdots \\ e_m \end{bmatrix} \tag{2}$$

where $l_n$ and $e_m$ denote the $n$-th type of output-energy source and the $m$-th type of input-energy, respectively, and $c_{mn}$ denotes the coupling factor.

We define the energy-flow distribution ratio from one energy-output port J (supply-side output or device output) to energy-input port K (demand-side input or device input) at different energy-transfer paths as the energy-flow distribution coefficient. The coupling coefficient can be expressed by the

energy-flow distribution coefficient, the energy-conversion efficiency of the device, and the associated state of the device and the load. The energy-conversion efficiency of the device is determined by the characteristics and operation situation of the device itself. The connection between the device and the load is the correspondence relationship between the energy converted by the conversion device and the load energy form. Then, the input–output relationship of the energy-conversion interface can be expressed by Equation (3).

$$\mathbf{L} = (\mathbf{F}^{\mathrm{T}}\mathbf{H}\mathbf{A}^{\mathrm{T}} + \mathbf{A}'^{\mathrm{T}}) \cdot \mathbf{E} \tag{3}$$

where $\mathbf{H}$ is the efficiency matrix and $h_{ii}$ represents the energy–conversion efficiency of the $i$-th device; $\mathbf{A}$ and $\mathbf{A}'$ are the distribution matrices; $\alpha_{ij}$ indicates the allocation ratio from $e_i$ to the $j$-th device; $\alpha'_{ij}$ indicates the allocation ratio from $e_i$ to the $j$-th load, and when there is one and only one energy-flow path, the value is 1; $\mathbf{F}$ is the branch connection characteristic matrix; $f_{ij}$ indicates the relationship of the $i$-th device and the $j$-th load, and its value is 1 when the branch exists. Otherwise, its value is 0.

### 2.2. Complex Input–Output Coupling Model

The energy flow of a complex ECI is shown in Figure 4. Compared to a simple ECI, there is an intermediate energy-flow process between the devices. The above simple input–output coupling model is no longer applicable. The input–output model for a complex ECI is derived below.

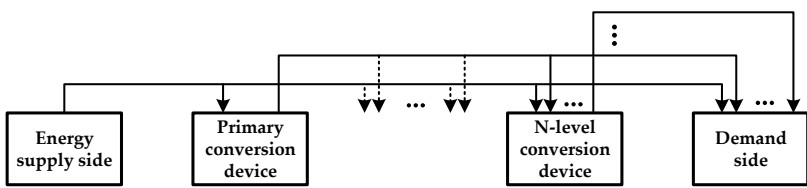

**Figure 4.** Energy-flow diagram of a complex energy-conversion interface.

For convenience of representation, this paper proposes a device-grading model. A device that directly receives energy from the energy-supply side and does not have another input from another device is defined as a primary conversion device. A device that is powered by a primary conversion device and does not have a higher-level input is a secondary conversion device. By this analogy, a device that is powered by an *S*-1 device and does not have a higher-level device is defined as an *S*-class conversion device.

In a complex ECI, there are *m* types of input energy, *n* types of output energy, *k* types of energy-conversion devices, and *S* energy-conversion levels, as shown in Figure 5. Different energy form collection other than the energy-supply side and the demand side occurs during the energy-transfer process. Assuming that there are *p* types of intermediate energy forms, the same energy forms existing on the energy-supply side and the demand side are treated as different energy forms. If *q* represents the number of same energy forms on both sides of supply and demand, there should be *q* virtual-conversion devices whose conversion factors is 1. Totally, there are *m* + *n* + *p* energy forms and *k* + *q* energy-conversion devices in the system.

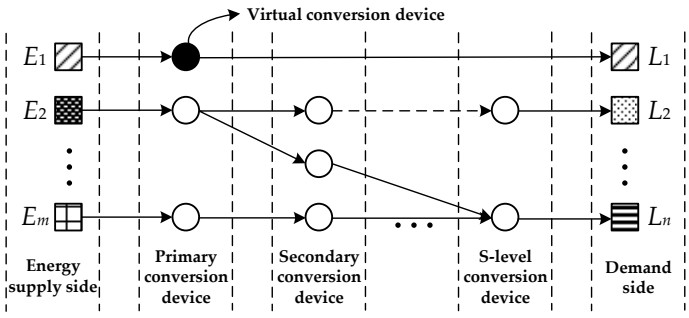

**Figure 5.** Schematic diagram of the hierarchical structure of a complex ECI.

Energy form property matrix **D** is defined to represent the correspondence between energy-output port J, energy-input port K, and the energy form in the system. In this matrix, $d_{ij}$ represents the correspondence between the $i$-th input/output port and the $j$-th energy form, that is, the value of $d_{ij}$ is 1 when the $i$-th port is about the $j$-th energy form, and 0 otherwise. **D1** ~ **D4** are the corresponding relationship between power-supply-side output port J1, device output port J2, demand-side input port K1, device input port K2, and the energy forms. The devices are arranged in order of level. According to the above complex model, the number of columns of the four matrices is $m + n + p$, and the numbers of rows of the four matrices are $m$, $\sum\limits_{i=1}^{k+q} N_i$, $n$, $\sum\limits_{i=1}^{k+q} M_i$ respectively, where $M_i$ and $N_i$ are the number of input ports and the number of output ports of the $i$-th energy-conversion device, respectively.

We define branch connection characteristic matrix **F** to indicate the connection relationship between each energy-output port and the energy-input port. Block matrices **F1** ~ **F4** represent the connection relationship between J1 and K1, J1 and K2, J2 and K1, J2 and K2, respectively. Then, the relationship between matrix **F** and energy form characteristic matrix **D** can be expressed as:

$$\mathbf{F} = \begin{bmatrix} \mathbf{D3} \cdot \mathbf{D1}^{\mathrm{T}} & \mathbf{D4} \cdot \mathbf{D1}^{\mathrm{T}} \\ \mathbf{D3} \cdot \mathbf{D2}^{\mathrm{T}} & \mathbf{D4} \cdot \mathbf{D2}^{\mathrm{T}} \end{bmatrix} = \begin{bmatrix} \mathbf{F1} & \mathbf{F2} \\ \mathbf{F3} & \mathbf{F4} \end{bmatrix} \tag{4}$$

The distribution matrix corresponds to the branch connection characteristic matrix. "1" in the branch connection characteristic matrices corresponds to the corresponding distribution coefficient, and "0" in the branch connection characteristic matrices corresponds to "0" in the corresponding distribution matrices. That is, the sparsity of the two characteristic matrices is the same, and the matrices are both highly sparse matrices. The distribution matrix satisfies Equation (5), that is, the sum of the elements of each column satisfying the distribution matrix is 1.

$$\sum A_j = 1 \tag{5}$$

The efficiency matrix of the device is defined as **H**, and $\mathbf{H_I}$ to $\mathbf{H_S}$ are the efficiency matrices from the primary conversion device to the $S$-level conversion device, respectively. The efficiency matrix is a generalized diagonal matrix. Using the matrix to represent its energy-transmission process requires multilevel calculation, and the branch relationship matrix is divided into $(s + 1) \times (s + 1)$ blocks according to device level and the relation between energy supply and demand sides, represented by $\mathbf{F}_{ij}$. The matrices are divided into generalized lower triangular matrices, as shown in Equation (6). For the convenience of calculation, the matrix was reblocked and represented by $\mathbf{F_I}$ to $\mathbf{F_{S+1}}$. $\mathbf{F_I}$ represents the correspondence between energy-supply side energy output and the first-order conversion device input, and $\mathbf{F_{II}}$ represents the correspondence between energy-supply side, the energy output of the primary conversion device, and the energy input of the secondary conversion device. $\mathbf{F_{S+1}}$ represents the correspondence between the energy outputs of the energy-supply side, the energy outputs of the primary conversion devices to the S conversion devices, and the energy inputs on the demand side. Therefore, block matrices $\mathbf{A_I}$ to $\mathbf{A_{S+1}}$ of the corresponding allocation matrix were obtained.

$$\mathbf{F} = \begin{bmatrix} \mathbf{F}_{11}^{\;\mathrm{I}} & \mathbf{0} & \cdots & \mathbf{0} \\ \mathbf{F}_{21} & \mathbf{F}_{22}^{\;\mathrm{II}} & \cdots & \mathbf{0} \\ \vdots & \vdots & \ddots & \vdots \\ \mathbf{F}_{s+1,1} & \mathbf{F}_{s+1,2} & \cdots & \mathbf{F}_{s+1,s+1}^{\;S+1} \end{bmatrix} \tag{6}$$

Finally, the input–output coupling model of the complex model was obtained as Equation (7), and its simplification is shown in Equation (8).

$$L = A_{S+1} \begin{bmatrix} E \\ H_I A_I E \\ \vdots \\ H_S A_S \begin{bmatrix} E \\ H_I A_I E \\ \vdots \\ H_{S-1} A_{S-1} \begin{bmatrix} E \\ H_I A_I E \\ \vdots \end{bmatrix} \end{bmatrix} \end{bmatrix} \tag{7}$$

$$L = (A_{S+1}^1 + A_{S+1}^2 H_I A_I \cdots + A_{S+1}^{S+1} H_S (A_S^1 + A_S^2 H_I A_I \cdots + A_S^S H_{S-1} (A_{S-1}^1 + \cdots))) E \tag{8}$$

## 3. Optimal Configuration Model of Energy-Conversion Interface

According to the evaluation indicators and configuration principles, an optimal configuration model is established in this section. It includes the relevant objective functions and constraints. It aims at improving the economic performance and energy-utilization level of the system. The optimal ECI configuration system is shown in Figure 6 [18,19].

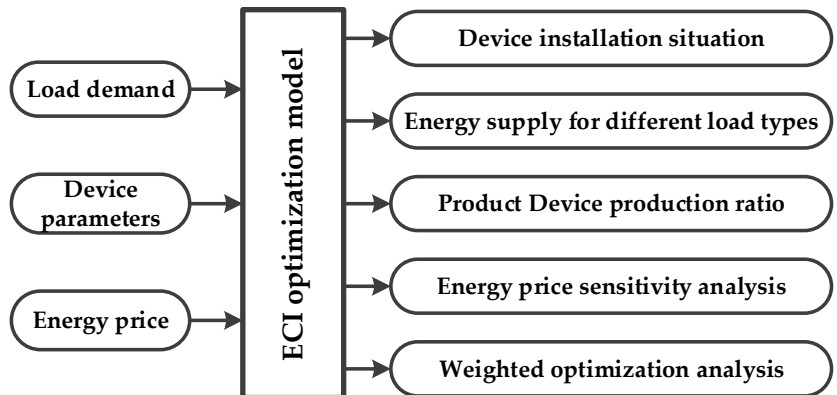

**Figure 6.** Optimal energy-conversion-interface configuration system.

### 3.1. Objective Function

#### 3.1.1. Economic Performance

As shown in Equations (9)–(14), aiming at minimizing the cost of the ECI, including energy purchase cost $C_1(t)$, device-installation cost $C_2$, device-maintenance cost $C_3(t)$, environmental costs $C_4(t)$, and electricity sales revenue $C_5(t)$, the following variables were optimized, including the type of installed energy-conversion device in the system, the corresponding installed capacity, and the energy-flow ratio at each moment.

Device-installation cost is calculated by the equivalent annual cost, which is the converted annual input cost considering life- and time-related market factors.

Device-maintenance cost is related to the working status and output level in each period.

Environmental cost is calculated by the environmental value converted from the pollutants discharged by the device during system operation [13].

$$C = \min\left(\sum C_1(t) + C_2 + \sum C_3(t) + \sum C_4(t) + \sum C_5(t)\right) \tag{9}$$

$$C_1(t) = \sum P_i(t)\lambda_i \tag{10}$$

$$C_2 = \sum \sigma_i \cdot N_i \cdot \frac{r(1+r)^{\tau_i}}{(1+r)^{\tau_i} - 1} \tag{11}$$

$$C_3(t) = \sum \delta_i \cdot P_i(t) \tag{12}$$

$$C_4(t) = \sum P_{ij}(t)\lambda_{ij}\gamma_{ij} \tag{13}$$

$$C_5(t) = P_s(t) \cdot c_s \tag{14}$$

Equation (10) is the energy purchase cost, which is obtained by multiplying the energy-consumption amount by the corresponding energy price. $\lambda_i$ is the unit price of the *i*-th energy.

Equation (11) is the device-installation cost, expressed by the equivalent annual cost. $\sigma_i$, $IC_i$, $\tau_i$ represent the unit installed price, quantity, and service life of the *i*-th device, respectively. $r$ is the discount rate.

Equation (12) is the device-maintenance cost, where $\delta_i$ is the maintenance cost corresponding to the unit energy output of the *i*-th device.

Equation (13) is the environmental cost. $\lambda_{ij}$ (¥/kg) is the environmental value of the *j*-th pollutant produced by the *i*-th device. $\gamma_{ij}$ (kg/kWh) is the intensity of the *j*-th class pollutant corresponding to the output unit energy of the device.

Equation (14) is the electricity sale revenue, $P_s(t)$ is the sale power, and $c_s$ is the sale price.

### 3.1.2. Energy-Utilization Efficiency

The overall energy-utilization efficiency of the system is one of the most important bases of evaluating IES structure rationality. There are many factors affecting energy loss in the transmission process, such as different energy-conversion efficiencies between several energy networks and different energy-transmission paths. Even if total energy demand is the same, the energy consumption of different energy forms may be still varied. Equation (15) is the energy-utilization efficiency of the IES, which can be calculated as the ratio of the total energy output and the total energy input. The total energy-output amount is shown in Equation (16). The total amount of energy input is shown in Equation (17).

$$\eta_{eu} = \max \frac{L_{sum}(t)}{E_{sum}(t)} \tag{15}$$

$$L_{sum}(t) = \sum_{i=1}^{n} L_i(t) \tag{16}$$

$$E_{sum}(t) = \sum_{i=1}^{m} E_i(t) \tag{17}$$

### *3.2. Restrictions*

Equations (18)–(22) are the model constraints that include energy-conservation, device-installation, and device-operation constraints.

### 3.2.1. Energy-Conservation Constraints

In the energy-conversion interface, energy at any node should be conservation. If the energy source does not have stored energy transposed in the system, it should satisfy Equation (18).

$$L_i(t) = \sum P_{i.in}(t) \tag{18}$$

where $L_i(t)$ is the *i*-th energy output, and $P_{i,in}(t)$ is the total power of the *i*-th energy source flowing into the energy output port.

If part of a certain energy form may be stored in the energy-storage device, it should satisfy Equation (19):

$$L_i(t) = \sum P_{i.in}(t) + P_{i.out}(t) \tag{19}$$

where $P_{i,out}(t)$ is the stored energy during the *t*-th period for the *i*-th energy source. When energy is stored, $P_{i,out}(t)$ is negative, and when it is discharged, $P_{i,out}(t)$ is positive.

### 3.2.2. Device-Installation Constraints

Device-installation capacity must meet the load demand and have a certain spare capacity at maximum load. At the same time, there should be a maximum-capacity limit according to actual installation conditions and device capacity. Therefore, the type and number of device installations should satisfy the inequality relationship in Equation (20):

$$\underline{S_i} \leq S_{i,t} \leq \overline{S_i} \tag{20}$$

where $S_i$ is the installation capacity of *i*-th device, $\overline{S_i}$ and $\underline{S_i}$ are the minimum and maximum limits of device-installation capacity.

### 3.2.3. Device-Operation Constraints

The device should meet its operating characteristics as shown in Equations (21) and (22) during the optimization process, including energy-conversion and device-output characteristics. Energy-conversion characteristics are shown in Equation (21). Device-output characteristics are shown in Equation (22). The output level of the device is related to the parameters of the device itself and fluctuates within the scope.

$$E_{o,j}(t) = H_j \cdot E_{i,j}(t) \tag{21}$$

$$\underline{P_j} \leq P_j(t) \leq \overline{P_j} \tag{22}$$

where $E_{o,j}(t)$ and $E_{i,j}(t)$ are the output and input of *j*-th device, $H_j$ is the output level of *j*-th device in the *t*-th period, $P_j(t)$ is the output level of *j*-th device in *t*-th period, and $\overline{P_j}$ and $\underline{P_j}$ are the maximum and minimum output of the *j*-th device. For energy-storage devices, $\overline{P_j}$ and $\underline{P_j}$ represent maximum and minimum energy storage and release power.

### 3.2.4. Energy-Dispatch Constraints

Energy conservation should be satisfied in the energy-dispatch process, and reflected in the dispatch factor as shown in Equation (8).

## 4. Case Analysis

A campus in the north of China is selected as the example. Its ECI planning model is shown in Figure 7.

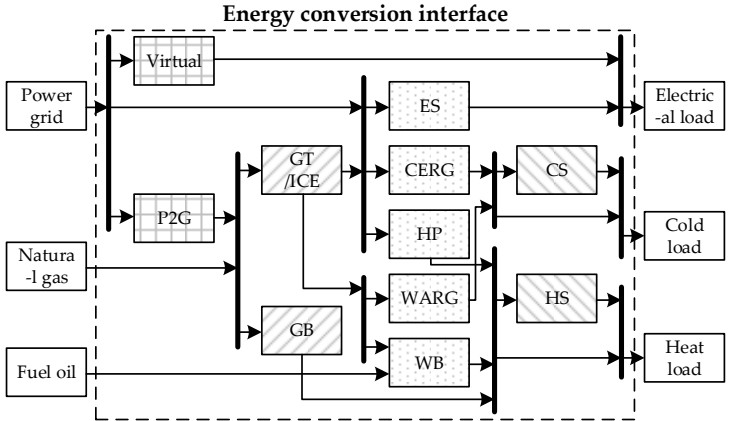

**Figure 7.** Schematic diagram of the optimal configuration of a campus ECI.

## 4.1. Campus Energy-Conversion-Interface Model

According to the energy-conversion modeling method described in Section 2.2, the P2G device is a primary conversion device, and the gas turbine (GT), internal combustion engine (ICE), and gas boiler (GB) are secondary conversion devices. The electric refrigerator (EC), absorption chiller (AC), electric-heat pump (HP), waste-heat boiler (WB), and electric storage (ES) are tertiary conversion devices. Cold storage (CS) and thermal storage (TS) are quaternary conversion devices. Due to the particularity of the energy-storage devices, the equation containing the remaining conversion devices is first deduced, and then the energy-storage device is specially treated. What should be pointed out is that, since power grid supply and electrical load are treated as two different forms of energy, it is assumed that there is a primary conversion device with an efficiency of 1 between power grid and electrical load.

The block matrix of energy-distribution matrix **A** corresponding to the branch connection characteristic matrix is shown in Equation (23) and satisfies the equality constraint of Equation (8).

$$
\begin{cases}
\mathbf{A_I} = \begin{bmatrix} \alpha_{11} & 0 & 0 \\ \alpha_{21} & 0 & 0 \end{bmatrix} \\[2mm]
\mathbf{A_{II}} = \begin{bmatrix} 0 & \alpha_{32} & 0 & 0 & \alpha_{35} \\ 0 & \alpha_{42} & 0 & 0 & \alpha_{45} \\ 0 & 0 & \alpha_{53} & 0 & 0 \end{bmatrix} \\[2mm]
\mathbf{A_{III}} = \begin{bmatrix} \alpha_{61} & 0 & 0 & 0 & 0 & \alpha_{66} & 0 & 0 \\ \alpha_{71} & 0 & 0 & 0 & 0 & \alpha_{76} & 0 & 0 \\ 0 & 0 & 0 & 0 & 0 & 0 & \alpha_{87} & 0 \\ 0 & 0 & 0 & 0 & 0 & 0 & \alpha_{97} & 0 \end{bmatrix} \\[2mm]
\mathbf{A_{IV}} = \begin{bmatrix} 0 & 0 & 0 & \alpha_{10,4} & 0 & 0 & 0 & 0 & 0 & 0 & 0 & 0 \\ 0 & 0 & 0 & 0 & 0 & 0 & 0 & 0 & \alpha_{11,9} & 0 & \alpha_{11,11} & 0 \\ 0 & 0 & 0 & 0 & 0 & 0 & 0 & \alpha_{12,8} & 0 & \alpha_{12,10} & 0 & \alpha_{12,12} \end{bmatrix}
\end{cases}
\tag{23}
$$

Mentioned herein are various types of gas turbines and internal-combustion engines. Assuming that there are *n* unit models, the operating characteristics of the gas turbine and internal-combustion engine can be expressed as:

$$
\begin{bmatrix} P_e \\ P_{th} \end{bmatrix} = \begin{bmatrix} \eta_{e,1} & \eta_{e,2} & \cdots & \eta_{e,n} \\ \eta_{th,1} & \eta_{th,2} & \cdots & \eta_{th,n} \end{bmatrix} \begin{bmatrix} \alpha_1 \\ \alpha_2 \\ \vdots \\ \alpha_n \end{bmatrix} E_{air} = \begin{bmatrix} \eta_e \\ \eta_{th} \end{bmatrix} E_{air}
\tag{24}
$$

The efficiency matrix of the energy-conversion devices at all levels is given by Equations (25)–(27).

$$\mathbf{H}_{\mathrm{I}} = \mathrm{diag}(1, \eta_{\mathrm{P2G}}) \tag{25}$$

$$\mathbf{H}_{\mathrm{II}} = \begin{bmatrix} \eta_e & 0 \\ \eta_{th} & 0 \\ 0 & \eta_{\mathrm{GB}} \end{bmatrix} \tag{26}$$

$$\mathbf{H}_{\mathrm{III}} = \mathrm{diag}(\eta_{\mathrm{EC}}, \eta_{\mathrm{HP}}, \eta_{\mathrm{AC}}, \eta_{\mathrm{WB}}) \tag{27}$$

Energy-storage devices are handled with the load in this paper. After passing through the energy-conversion device, when energy appears in a form corresponding to the load, there are two energy-transmission paths, direct supply to load or storage. The relationship between energy input and load can be shown as Equation (28), and expressed in matrix form as Equation (29):

$$L_i = (1 - X_i \alpha_{s,i}) E_{in,i} + Y_i \beta_{s,i} E_{s,i} \tag{28}$$

$$\mathbf{L} = (1 - \mathbf{X}\mathbf{A_s})\mathbf{E_{in}} + \mathbf{Y}\mathbf{B_s}\mathbf{E_s} \tag{29}$$

The values of $X_i$ and $Y_i$ may be 0 or 1, satisfying $X_i + Y_i = 1$. Matrix $\mathbf{X} = \mathrm{diag}(X_1, X_2, \ldots X_n)$, $\mathbf{Y} = \mathrm{diag}(Y_1, Y_2, \ldots Y_n)$, where $n$ is the number of energy forms of the load. $\alpha_{s,i}$ and $\beta_{s,i}$ are, respectively, energy-storage and energy-release dispatch factors corresponding to the $i$-th energy-storage form according to matrices $\mathbf{A_s} = (\alpha_{s,1}, \alpha_{s,2}, \ldots \alpha_{s,n})^T$ and $\mathbf{B_s} = (\beta_{s,1}, \beta_{s,2}, \ldots \beta_{s,n})^T$. $E_{in,I}$ and $E_{s,i}$ are the energy input after energy passes through the energy-conversion device, and the energy stored in the energy-storage device, respectively, corresponding to two column vectors, $\mathbf{E_{in}}$ and $\mathbf{E_s}$, whose length is $n$.

The input–output coupling model of the complicated model is finally obtained as shown in Equation (30):

$$\begin{aligned} \mathbf{L} = \quad & (1 - \mathbf{X}\mathbf{A_s})[\mathbf{A}_{\mathrm{IV}}^1 + \mathbf{A}_{\mathrm{IV}}^2 \mathbf{H}_{\mathrm{I}} \mathbf{A}_{\mathrm{I}} + \mathbf{A}_{\mathrm{IV}}^3 \mathbf{H}_{\mathrm{II}}(\mathbf{A}_{\mathrm{II}}^1 + \mathbf{A}_{\mathrm{II}}^2 \mathbf{H}_{\mathrm{I}} \mathbf{A}_{\mathrm{I}}) + \\ & \mathbf{A}_{\mathrm{IV}}^4 \mathbf{H}_{\mathrm{III}}(\mathbf{A}_{\mathrm{III}}^1 + \mathbf{A}_{\mathrm{III}}^2 \mathbf{H}_{\mathrm{I}} \mathbf{A}_{\mathrm{I}} + \mathbf{A}_{\mathrm{III}}^3 \mathbf{H}_{\mathrm{II}}(\mathbf{A}_{\mathrm{II}}^1 + \mathbf{A}_{\mathrm{II}}^2 \mathbf{H}_{\mathrm{I}} \mathbf{A}_{\mathrm{I}}))]\mathbf{E} + \mathbf{Y}\mathbf{B_s}\mathbf{E_s} \end{aligned} \tag{30}$$

### 4.2. Case Data

The research period of the campus load was one year, which was divided into twelve time periods, that is, twelve months. In order to simplify the calculation, all months were classified into three categories according to seasonal characteristics. The load fluctuation of each seasonal characteristic is represented by load fluctuations of typical days. December, January, and February were classified as typical loads in winter. June to August were classified as typical loads in summer, and the other months are typical loads in spring and autumn. The maximum electrical, heat, and cold load for each month is shown in Figure 8. The load of a typical day of each season and the ratio of the maximum load in those days are shown in Figure 9.

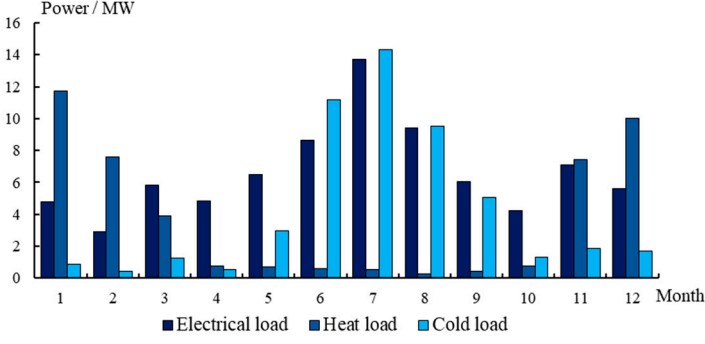

**Figure 8.** Maximum load distribution for each month of the year.

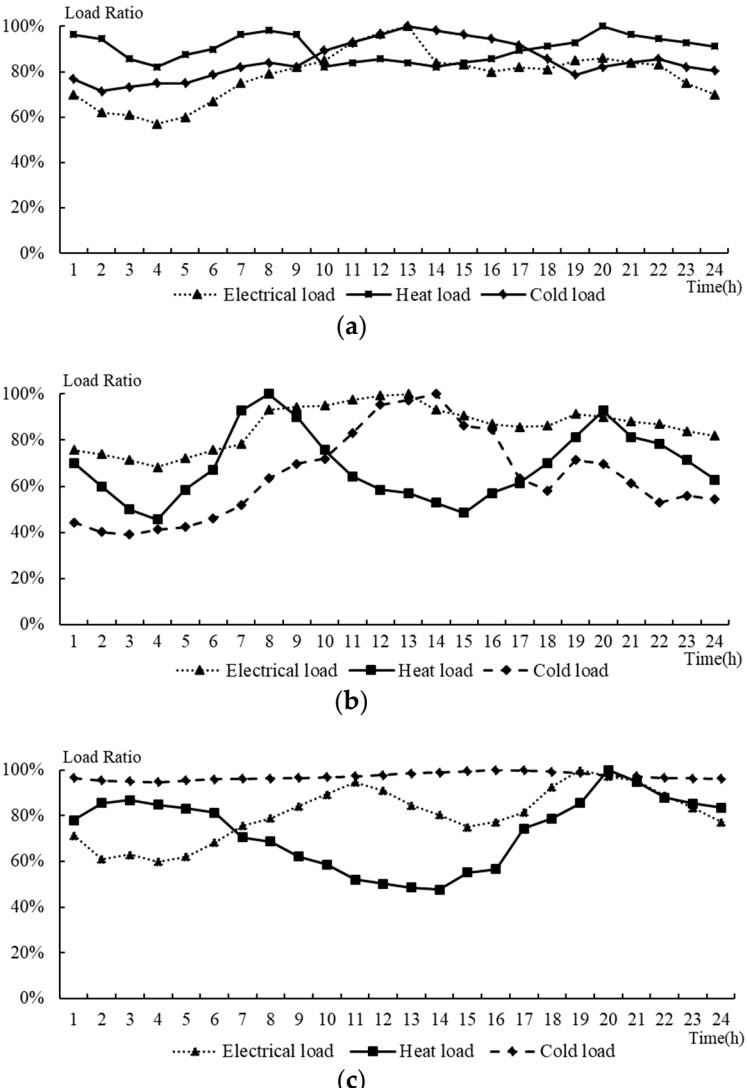

**Figure 9.** Load ratio at each moment in a typical day to maximum load (**a**) in spring and autumn; (**b**) in summer; (**c**) in winter.

The literature [14] proposed the particle swarm optimization (PSO) algorithm with inertia weight, which is applied to the solution of the example. The basic parameters of the PSO algorithm were set as follows: number of particles, 50; acceleration factors $c_1$ and $c_2$, 1.494; $w_{start} = 0.95$ and $w_{end} = 0.4$; take control factor $d_1 = 0.2$, $d_2 = 7$, and maximum number of iterations, 10,000. The PSO flowchart is shown in Figure A1. The parameters of each device and related system parameters are shown in Tables A1 and A2 [19–28]. Taking whether the system device will be installed, installation quantity, and energy-flow dispatch as optimization variates, the campus is optimized for the purposes of economy and energy-utilization efficiency.

*4.3. Result Analysis*

4.3.1. Configuration Result

The PSO algorithm described above is used to optimize system economy and energy efficiency. By programming and calculating the case data in MATLAB, the number of installed devices was obtained and is shown in Table 1. It can be seen from the table that, when choosing economy as the optimization goal, the total cost of the system is reduced by 21.13% compared with the cost of

choosing energy-utilization efficiency as the goal, and energy-utilization efficiency is 45.46% lower than energy efficiency.

**Table 1.** Number of installed devices under different objective functions.

| Aims | Number of Installed Devices | | | | | | | | | | | Target Value | |
| --- | --- | --- | --- | --- | --- | --- | --- | --- | --- | --- | --- | --- | --- |
| | GT | ICE | P2G | WB | HP | GB | EC | AC | ES | TS | CS | *C* (Ten Thousand Yuan) | $\eta_{eu}$ |
| **Economy** | 1(4#) | 0 | 0 | 7 | 2 | 0 | 2 | 4 | 309 | 62 | 67 | 4385.18 | 0.853 |
| | 0 | 0 | 0 | 0 | 6 | 0 | 15 | 0 | 0 | 0 | 0 | 5560.28 | 1.564 |

Economic Analysis

It can be seen from Table 1 that from the obtained device-configuration result from the minimum comprehensive system cost, a gas turbine with higher power-generation efficiency was selected and installed. For the electricity–gas conversion device, energy consumption exists in the process of electric-energy conversion, and gas supply was sufficient in this paper, so it was not installed. The gas boiler was not installed, either, due to the existence of a highly efficient electric-heat pump. The remaining devices were installed because of their own advantages. For example, the input energy of absorption refrigeration and the waste-heat boiler is the residual smoke of the device with lower cost, which can improve the efficiency of the entire energy-production device; electric refrigerators and electric-heat pumps have the characteristic of high energy-conversion efficiency. The maximum output level of each device in different months of the year is shown in Figure 10. Figure 11 shows the annual energy-flow output of each device in the energy-conversion interface. The coupling of various energy forms can also be seen in Figure 11.

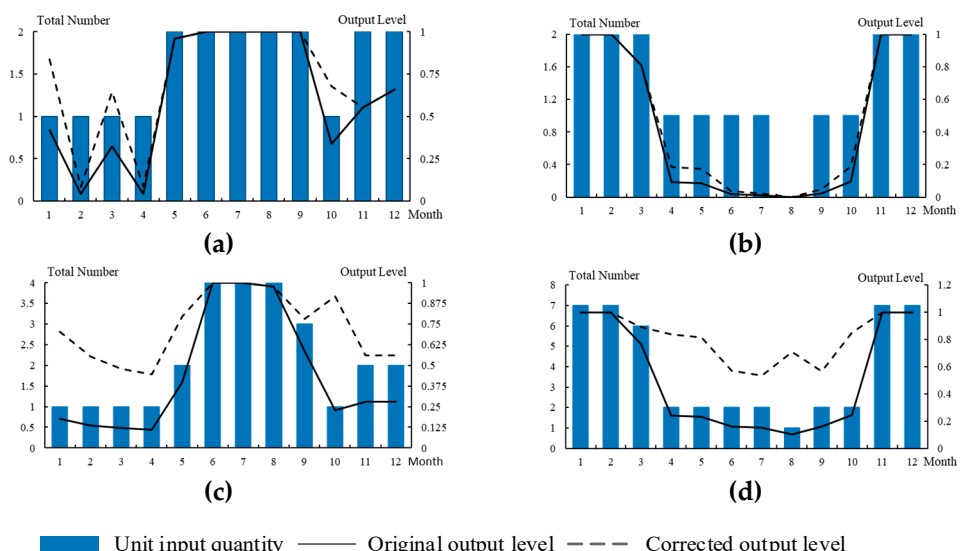

**Figure 10.** Device output level and unit adjustment: (**a**) electric refrigerator; (**b**) electric-heat pump; (**c**) absorption chiller; (**d**) waste-heat boiler.

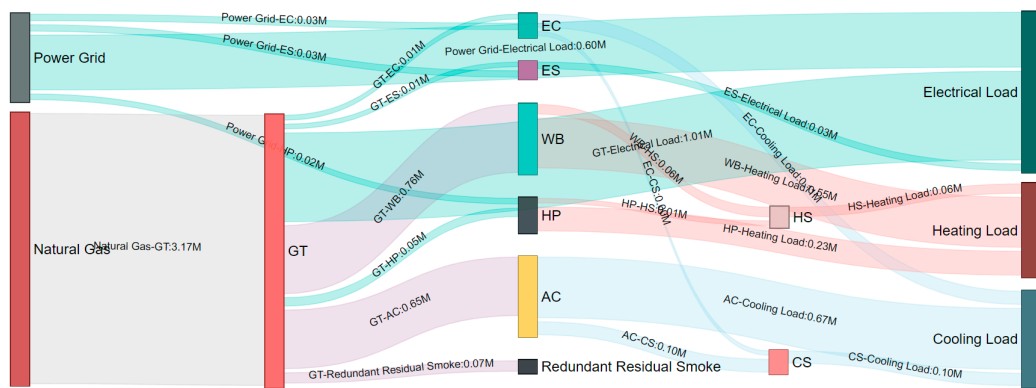

**Figure 11.** Annual energy-flow output of each device in the system.

Figure 12 shows the output of the device with daily running power conservation, and gives the graphs (Figure 13) of electricity conservation on the typical days of January, July, and October. The electric-load supply consists of power-grid purchase, gas-turbine power generation, and electric-energy storage discharge. In addition to electric-load supply, there are two power-flow directions of electricity storage and electricity sales.

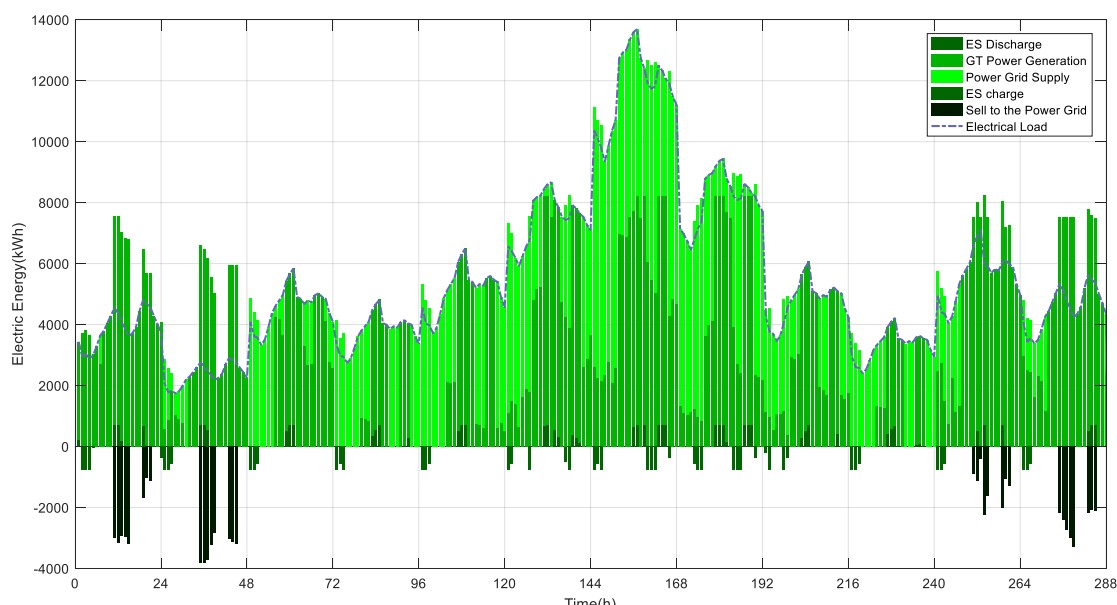

**Figure 12.** Output situation of daily running power-conservation device.

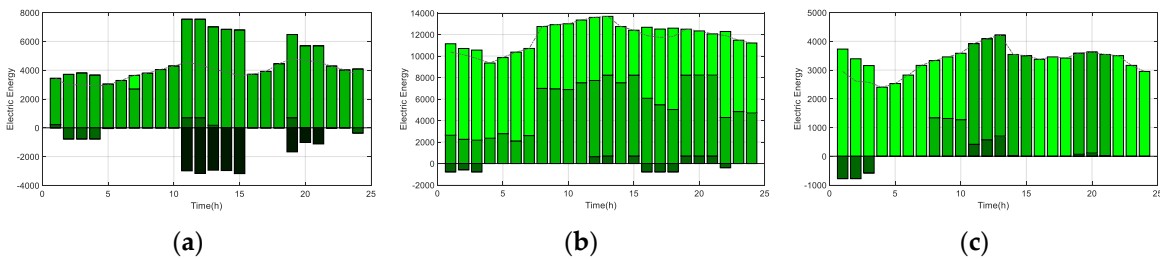

**Figure 13.** Device output with constant current on a typical season: (**a**) January; (**b**) July; (**c**) October.

As shown from the above figures, the overall system energy flow satisfies the conservation of electrical energy. In the month when electric load was relatively low, gas-turbine power generation mainly met the load demand, and electricity was sold at the peak of the electricity price. In the month when electric load was high, purchased electricity in the system was large, so there were no electricity

sales. It can be seen from the device output of the typical season that electric-load supply mainly came from grid-power supply and gas-turbine power generation. In the low-price stage of electricity, a small amount of energy storage occurred due to a lower energy cost. In the peak stage of electricity price, due to the high cost of electricity, the energy storage device will release the energy and sell the remaining electricity generated by the gas turbine online, thereby improving the economy of the system.

Figure 14 shows the output of the device with daily heat-energy conservation, and gives the heat-energy conservation map for three seasons. Heat-load supply consists of a waste-heat boiler and an electric-heat pump. In addition to heat-load supply, there is thermal-energy storage.

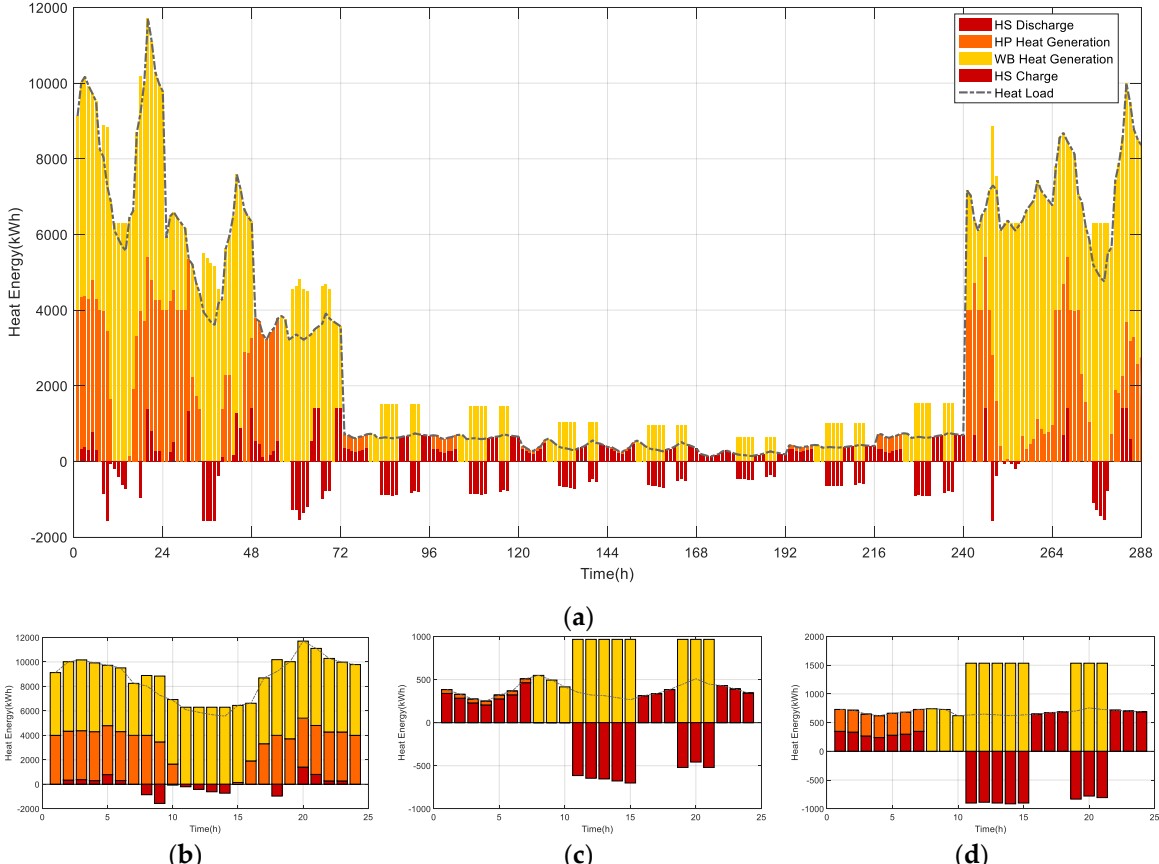

**Figure 14.** Daily running of heat-conserving device: (**a**) days of running heat-conservation device output; (**b**) January; (**c**) July; (**d**) October.

As can be seen from Figure 14, the whole system satisfies the conservation of thermal energy. The overall change level of the thermal load in one year is opposite to the change trend of the electric load and cold load in the system. In a month when the heat load is high, the cold load and the electric load are relatively low. The residual smoke generated by the gas turbine is fully used by the waste-heat boiler. The residual smoke can be regarded as zero-cost energy. The heat supply is preferentially supplied by the waste-heat boiler. When the waste-heat boiler cannot meet the heat-load demand, the higher-efficiency electric-heat pump is used to improve the thermal efficiency of the system. It can be seen from the heat-conservation figure of a typical season that heat energy is mainly supplied by the waste-heat boiler and the electric-heat pump. Heat energy is reserved when the heat of the waste-heat boiler is abundant, and the electric-heat pump is mainly applied during the lower-price period.

Figure 15 shows the output of the device with daily cooling energy conservation, and gives the cooling energy conservation graph for typical days of three seasons. The cold-load supply consists

of an electric chiller and an absorption chiller. In addition to the cold-load supply, there is also cold-energy storage.

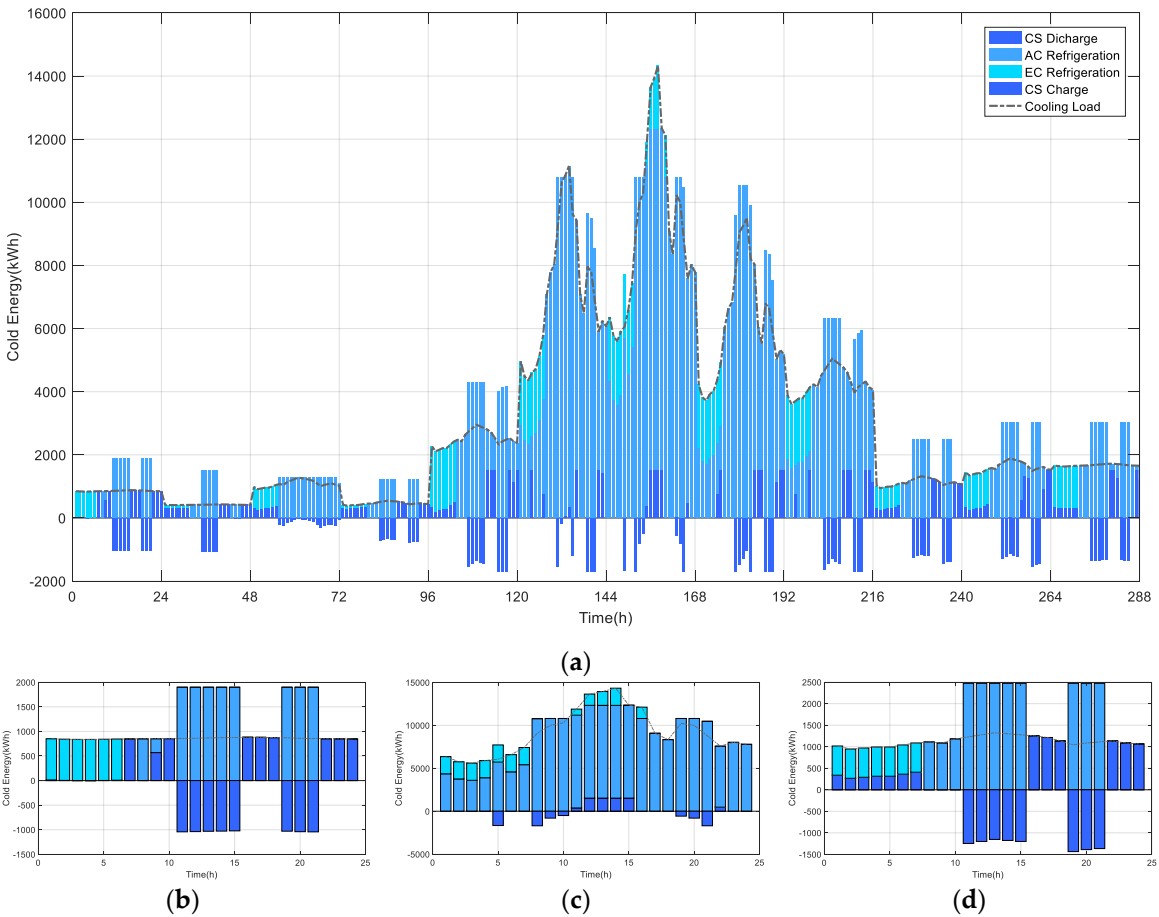

**Figure 15.** Daily output of cold-conserved device: (**a**) days of running cold-conservation device output; (**b**) January; (**c**) July; (**d**) October.

As concluded from the figure above, the whole system satisfies the conservation of cold energy. The change law of the cold load is positively correlated with the electric load. Therefore, residual smoke generated by the gas turbine can be fully utilized by the absorption chiller during the high electric-load period to improve the cooling efficiency of the system. When the absorption chiller cannot meet refrigeration demand, the electric chiller is applied. This can be obtained from the cold-conservation graph of the typical season. The system-cooling load of the system is mainly supplied by the absorption chiller. When the working efficiency of the absorption chiller is relatively high, the cold-energy storage device stores cold energy, and the cold energy is released when absorption refrigeration fails to meet cold-load demand. The electric refrigerator has high energy-conversion efficiency, but because of its high energy cost, as well as its restricted outputs, this only occurs during the period of low electricity price so as to meet the energy conservation of the system.

Energy-Use Efficiency Analysis

From Table 1 we can see when the comprehensive energy-utilization efficiency of the system is optimized. Overall energy utilization has efficiency over 1.0 due to the high conversion efficiency of electric refrigeration and the electric-heat pump. As per the results of device configuration, only the electric refrigerator and the electric-heat pump with a high energy-utilization rate could be installed. The maximum output level of each device in different months of the year is shown in Figure 16.

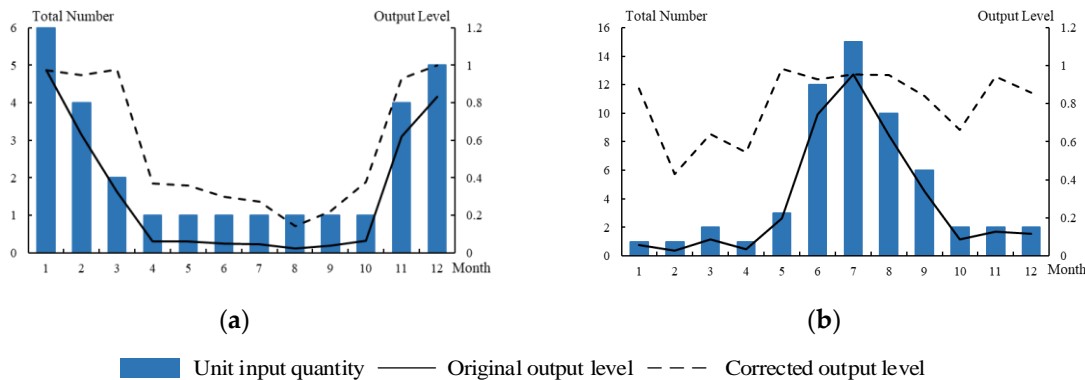

**Figure 16.** Device output level and unit adjustment: (**a**) electric-heat pump; (**b**) electric refrigerator.

As can be seen from Figure 16, the output level of the device has a large-scale fluctuation range. The load rate of several months is below 0.2, so the load rate of the device is low. By adjusting the actual working quantity of the unit, the corrected output level of the device is obviously improved. The reasonable option among the devices could also relatively improve the service life of the devices.

The energy-flow figure obtained with the goal of energy efficiency is relatively simple. As shown in Figure 17, daily-operation power distribution notes that electricity purchased from the grid flows to the electric load, electric-heat pump, and electric refrigerator. It can be seen from the figure that the purchase of electricity is mainly for electric-load supply. Since the conversion efficiency of the electric-heat pump and the electric refrigerator is high, only a small amount of electric energy input into the two devices can meet cooling-load and heating-load system demands.

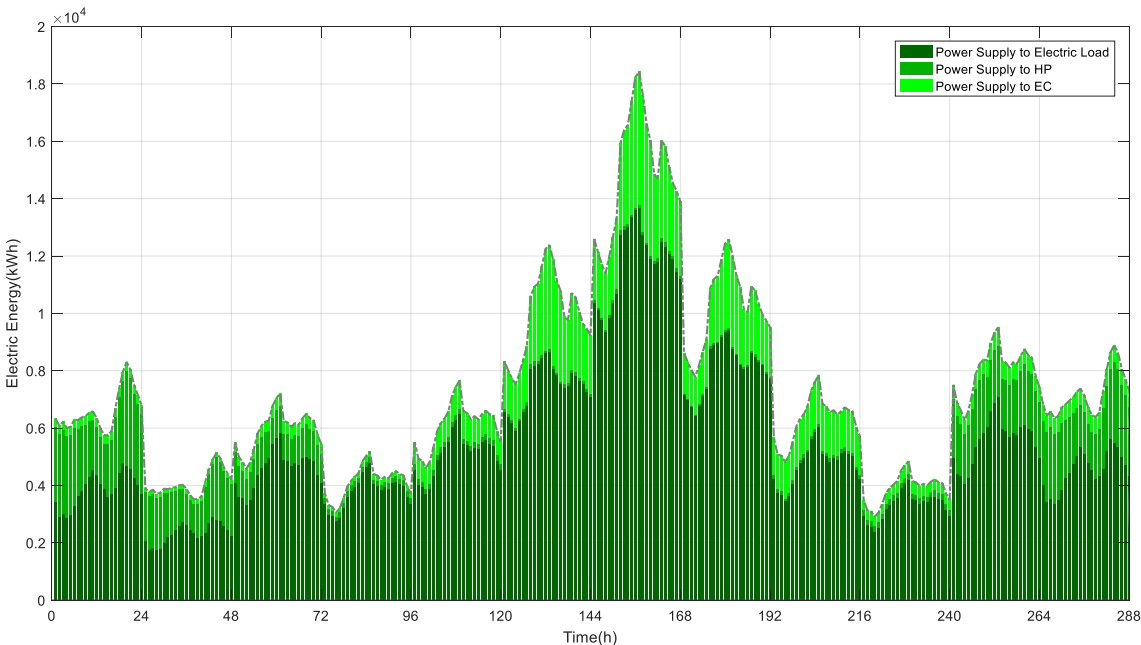

**Figure 17.** Diagram of daily-operation power distribution.

### 4.3.2. Sensitivity Analysis

The sensitivity of the optimization model indicates the sensitivity of the optimization targets to relevant system-parameter changes. It may be used to observe the adaptability of the optimization results to system-parameter changes. The smaller the sensitivity value is, the stronger the adaptability of the optimization results to parameter changes. Since the increase of device energy-conversion efficiency is slow due to technological developments, this paper only considers the adaptability of

economic system-optimization results to energy-cost changes. Figure 18 shows the impact of energy price fluctuation on system cost. The abscissa represents the ratio of energy price to its original value, and the ordinate represents the variation amplitude of the system economic indicator to its optimal value.

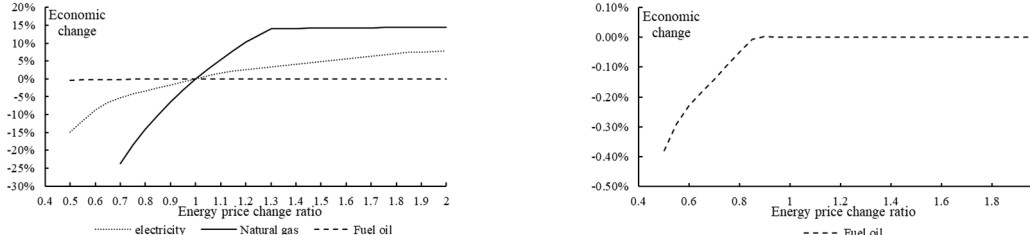

**Figure 18.** System economic sensitivity to energy prices.

As can be seen from Figure 18, the total cost of the system shows a general upward trend with the increase of energy costs. The lower the energy price, the greater the system-cost sensitivity to energy price changes. As the energy price increases, the total cost of the system gradually decreases, because the cost of purchasing energy from the system becomes larger. From the overall results, the price of natural gas has the greatest impact on the economy of the system, which then follows the price of electricity purchase. The price of fuel has the least impact on the system economy.

### 4.3.3. Weighted System Optimization

According to the optimal economic target value and the optimal energy-utilization efficiency value, the proportion of economy and energy-utilization efficiency is changed to optimize the system. The specific target value is shown in Equation (31), where $w_1$ and $w_2$ are weights, and the sum of the two is 1.0. The two target values and the number of devices installed with different weights were obtained. Figure 19 shows the two target values and the number of devices installed under different weights. Figure 20 shows the installation of devices under different proportions.

$$f = \min(w_1 \frac{C - C_{\min}}{C_{\min}} + w_2 \frac{\eta_{eu,\max} - \eta_{eu}}{\eta_{eu,\max}}) \tag{31}$$

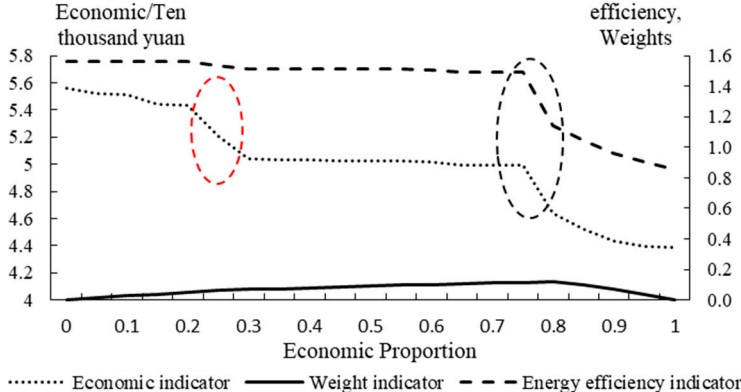

**Figure 19.** Optimization target values under different weights.

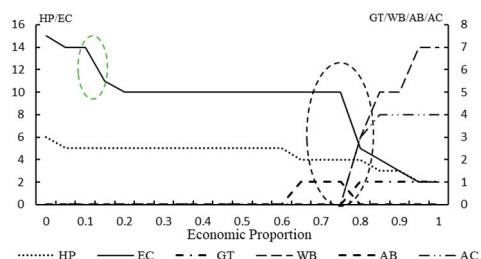 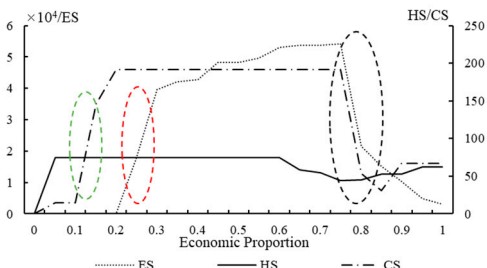

**Figure 20.** Device installation under different proportions.

As can be seen from Figure 20, as the economy proportion increases, economy optimization is better and energy-use efficiency is reduced. Therein, economic and energy-efficiency indicators have relatively large fluctuations under a certain proportion (circles in Figure 19). The reason is changes in different device types in the system. Certain devices appearing or disappearing can influence the plan of other devices, resulting in greatly changing the optimal target.

## 5. Conclusions

Building an efficient IES greatly promotes energy-utilization efficiency and improves system economy. This paper focuses on two difficulties: (1) initial IES planning that grows out of nothing and (2) joint planning of IES energy dispatch optimization and device planning.

The paper proposes an innovative energy-conversion interface model and simplifies a terminal IES into a multi-input to multioutput dual-port network. First, according to graph theory, the topological layering of system devices was carried out, the mathematical relationship of the energy input–output coupling was derived, and an overall IES model was obtained. Second, considering the IES dispatch optimization problem, an optimized ECI configuration model was established with the goal of system economy and energy utilization efficiency. Finally, the IES of a typical area was modeled and analyzed with the particle-swarm optimization algorithm to optimize the configuration to improve the economy and energy-utilization levels. The installed capacity of devices under different optimization targets and device output level at each moment was obtained. The energy-flow directions of the system devices were analyzed. Sensitivity analysis shows that system costs are the most sensitive to energy prices. For example, natural-gas price has the greatest impact on system economy in the study case. Weighted system optimization shows that as the economy proportion increases, the economy of the optimization result is better, and energy-utilization efficiency is reduced. This provides a reference for decision making on IES planning.

China has already started the construction of an energy Internet at a regional level. The National Energy Administration announced the first batch of smart-energy (energy Internet) demonstration projects in June 2017. The planning of regional IES has become the primary problem of these demonstration projects. The proposed model in this paper has higher practicability and ductility. It can be extended to more complex systems to facilitate energy-flow analysis, and is helpful to the IES plan.

**Author Contributions:** Y.Z., N.Z., and Z.Y. designed the study; X.Y. collected the simulation data; Z.Y. and N.Z. carried out the simulation and analyses; X.Y. and L.Z. wrote the manuscript; and R.X. and Y.Z. reviewed and edited the manuscript.

**Conflicts of Interest:** The authors declare no conflict of interest.

## Appendix A

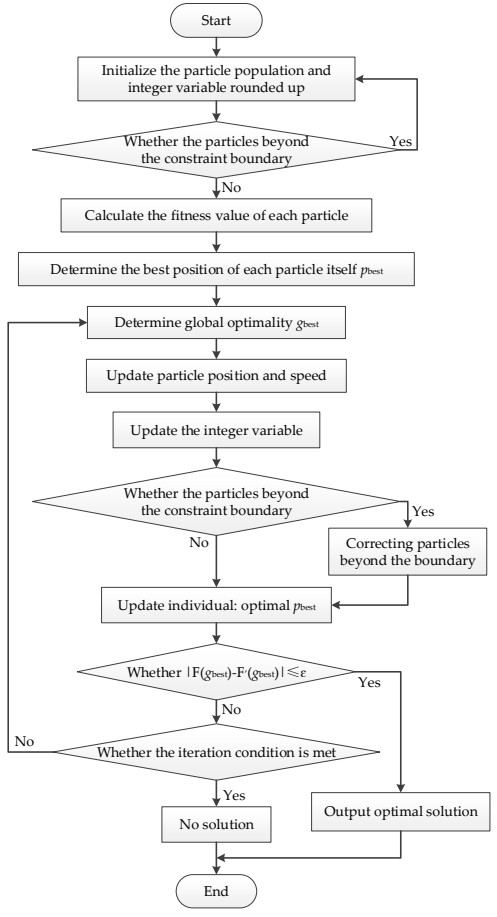

**Figure A1.** Particle swarm optimization flowchart.

## Appendix B

**Table A1.** Device parameters.

| Type | | Output Rating (kW) | | Conversion Coefficient | | Investment Cost (Yuan/kW) | Operating and Maintenance Cost (Yuan/kW) | Durable Years |
|---|---|---|---|---|---|---|---|---|
| | | Power | Thermal | Power | Thermal | | | |
| GT | 1 | 3500 | 6000 | 0.28 | 0.48 | 15,428,000 | | |
| | 2 | 4600 | 7800 | 0.30 | 0.50 | 19,710,000 | 0.06 | 30 |
| | 3 | 5200 | 8300 | 0.30 | 0.48 | 21,931,520 | | |
| | 4 | 7520 | 10,400 | 0.34 | 0.47 | 29,762,000 | | |
| ICE | 1 | 800 | 1460 | 0.15 | 0.28 | 4,422,400 | 0.06 | 30 |
| | 2 | 534 | 780 | 0.15 | 0.22 | 3,336,000 | | |
| WB | | 900 | | 0.8 | | 600,000 | 0.003 | 15 |
| HP | | 2000 | | 3.5 | | 6,000,000 | 0.007 | 20 |
| GB | | 3100 | | 0.9 | | 2,480,000 | 0.004 | 20 |
| EC | | 1000 | | 3 | | 1,000,000 | 0.0097 | 15 |
| AC | | 2700 | | 1.2 | | 4,968,000 | 0.008 | 30 |
| P2G | | 1000 | | 0.5 | | 4,800,000 | / | 8 |
| ES | | 12 | | 0.95 | | 4,000 | / | 5 |
| TS | | 120 | | 0.95 | | 24,480 | / | 15 |
| CS | | 120 | | 0.95 | | 24,000 | / | 15 |

**Table A2.** Energy price.

| Electricity Price (Yuan/kWh) | | Natural Gas Price (Yuan/m$^3$) | Fuel Oil Price (Yuan/kg) |
|---|---|---|---|
| Purchase | Sale | | |
| 0.83 (9–13, 17–19 h) 0.49 (8–10, 14–16, 20–22 h) 0.17 (1–7 h) | 0.65 (9–13, 17–19 h) 0.38 (8–10, 14–16, 20–22 h) 0.13 (1–7 h) | 2.67 | 2.2 |

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
