# Peer review of "Optimal Configuration of Integrated Energy System Based on Energy-Conversion Interface"

_applsci, doi:10.3390/app9071367_

Round 1
Reviewer 1 Report
Integrated Energy System Optimization Configuration Based on Energy Conversion Interface
Zicong Yu 1,*, Xiaohua Yang 1, Lu Zhang 1 ,Yongqiang Zhu 1, Ruihua Xia 1 and Na Zhao 1
This paper proposes an energy conversion interface (ECI) model which abstracted the complex multi-energy network into a multi-input-multi-output dual-port network.
The paper is well written, the model is clearly explained, and the test case (the park sketched in Fig. 7) is convincing and well explained.
Considering equipment installation cost, operation and maintenance cost is a strong feature, because such things are often neglected.
I have some comments
1)
The proposed model seems to be static (cf. (1) and (2)). This in my impression excludes any form of storage which are dynamics because they can accumulate and release power. Is my understading correct? Please clarify
2)
In practise the conversion coefficients might be unknown or change with time (e.g. due to loss of efficiency which require maintenance). How can you cope with this situation?
3)
solving the model is done via particle swarm optimization method. I would like to see a learning curve of such algorithms, to see how many iterations are necessary for convergence to the optimal solution
4)
The energy interface (also known as energy hub) has been intensively studied recently and use in smart grids/ microgrids
Distributed MPC for frequency regulation in multi-terminal HVDC grids
Control Engineering Practice, Volume 46, January 2016, Pages 176-187
Paul Mc Namara, Rudy R. Negenborn, Bart De Schutter, Gordon Lightbody, Seán McLoone
Smart distribution system management considering electrical and thermal demand response of energy hubs
Energy, Volume 169, 15 February 2019, Pages 38-49
Vahid Davatgaran, Mohsen Saniei, Seyed Saeidollah Mortazavi
Grid-Connected Microgrids: Demand Management via Distributed Control and Human-in-the-Loop Optimization
Advances in Renewable Energies and Power Technologies, 2018, Pages 315-344
Christos D. Korkas, Simone Baldi, Elias B. Kosmatopoulos
Optimal bidding strategy for an energy hub in energy market
Energy, Volume 148, 1 April 2018, Pages 482-493
Vahid Davatgaran, Mohsen Saniei, Seyed Saeidollah Mortazavi
Occupancy-based demand response and thermal comfort optimization in microgrids with renewable energy sources and energy storage
Applied Energy, Volume 163, 1 February 2016, Pages 93-104
Christos D. Korkas, Simone Baldi, Iakovos Michailidis, Elias B. Kosmatopoulos
I feel that all these references are missing and are somehow quite important (apearing in the top journals of the field)
Author Response
Response to Reviewer 1 Comments
I am very grateful to your comments for the manuscript. According with your advice, we amended the relevant part in manuscript. We sincerely hope this manuscript will be finally acceptable to be published. The point to point responds to the comments are listed as following:
Comment 1: The proposed model seems to be static (cf. (1) and (2)). This in my impression excludes any form of storage which are dynamics because they can accumulate and release power. Is my understanding correct? Please clarify.
Response 1: We are very sorry that the misunderstanding has been caused by the unclear expression of the original manuscript. The optimized configuration model proposed in this paper is dynamic. Equation (1)-(2) or Equation (8) is the energy coupling relationship between the input side and the output side, which is a constraint of the proposed model. The energy storage model of this paper is dynamic, see equation (19). We have made adjustments to the manuscript to make it clearer.
Comment 2: In practice, the conversion coefficients might be unknown or change with time (e.g. due to loss of efficiency which require maintenance). How can you cope with this situation?
Response 2: Thanks for the reviewer’s kindly suggestion. In order to simplify the model, the conversion coefficients of the equipment in this paper is fixed, which is our shortcoming. In future work, we will take the uncertainties in the system into consideration, which we mentioned at the end of the revised manuscript.
Comment 3: Solving the model is done via particle swarm optimization method. I would like to see a learning curve of such algorithms, to see how many iterations are necessary for convergence to the optimal solution.
Response 3: Due to space limitations, we did not explain too much about the particle swarm algorithm in the original manuscript. Thanks for your interest in the algorithm. We give the iteration curve in the upload PDF.
Comment 4: The energy interface (also known as energy hub) has been intensively studied recently and use in smart grids/ microgrids.
Distributed MPC for frequency regulation in multi-terminal HVDC grids
Control Engineering Practice, Volume 46, January 2016, Pages 176-187
Paul Mc Namara, Rudy R. Negenborn, Bart De Schutter, Gordon Lightbody, Seán McLoone
Smart distribution system management considering electrical and thermal demand response of energy hubs
Energy, Volume 169, 15 February 2019, Pages 38-49
Vahid Davatgaran, Mohsen Saniei, Seyed Saeidollah Mortazavi
Grid-Connected Microgrids: Demand Management via Distributed Control and Human-in-the-Loop Optimization
Advances in Renewable Energies and Power Technologies, 2018, Pages 315-344
Christos D. Korkas, Simone Baldi, Elias B. Kosmatopoulos
Optimal bidding strategy for an energy hub in energy market
Energy, Volume 148, 1 April 2018, Pages 482-493
Vahid Davatgaran, Mohsen Saniei, Seyed Saeidollah Mortazavi
Occupancy-based demand response and thermal comfort optimization in microgrids with renewable energy sources and energy storage
Applied Energy, Volume 163, 1 February 2016, Pages 93-104
Christos D. Korkas, Simone Baldi, Iakovos Michailidis, Elias B. Kosmatopoulos
I feel that all these references are missing and are somehow quite important (appearing in the top journals of the field).
Response 4: Thanks for the literatures, which we find very valuable and has been added to the references in this article.

Reviewer 2 Report
In this paper, the authors have presented an ECI model which can achieve the energy coupling between the energy supply side and the demand side. For better clarification, the authors are advised to address the following questions/concerns:
1) The authors may wish to report in more details of their research; the current paper does not show enough research contributions in relation to previous work.
2) The reviewer thinks it is important to compare with other methods proposed in the literature, highlighting the advantages and disadvantages of the proposed method.
3) Overall, the writing and presentation of the paper are acceptable but the reviewer found some minor grammatical errors and typos. Please carefully proof-read the paper and correct them.
Author Response
Response to Reviewer 2 Comments
I am very grateful to your comments for the manuscript. According with your advice, we amended the relevant part in manuscript. We sincerely hope this manuscript will be finally acceptable to be published. The point to point responds to the comments are listed as following:
Comment 1: The authors may wish to report in more details of their research; the current paper does not show enough research contributions in relation to previous work.
Response 1: We are very sorry that this part was not clear in the original manuscript.
At present, the optimization problems of regional integrated energy systems can be divided into two categories: 1) optimizing the capacity and model of the devices for a given regional IES; 2) optimizing the energy flow for existing system topologies. After reading a lot of literature, we found there are two difficulties in the study of IES: 1) simultaneous planning the topology and device capacity or model of the IES, that is, the planning of the IES starts from scratch; 2) combining the energy dispatch optimization of the IES with the planning of the devices to support each other.
Firstly, this paper establishes an energy input-output coupling model by deducing the energy input-output relationship of the energy conversion interface and classifying and refining the energy conversion devices. The model has good ductility and makes the relationship between energy flow and devices in the system become clear. Secondly, on the basis of economic indicators such as device installation cost, operation and maintenance cost, and annual cost, as well as energy-saving indicators such as energy utilization efficiency, an optimized configuration model for ECI is established. The model has the following innovations: 1) the planning of the IES starts from scratch; 2) energy optimization dispatch and device quantity are planned jointly.
We added this point into our revised manuscript.
Comment 2: The reviewer thinks it is important to compare with other methods proposed in the literature, highlighting the advantages and disadvantages of the proposed method.
Response 2: We are very sorry that this part was not clear in the original manuscript.
The existing related research has two difficulties mentioned above. In view of the above difficulties, this paper proposes an innovative energy conversion interface model (ECI), simplifies terminal IES into a multi-input to multi-output dual-port network. The advantage of this model is that simultaneous planning the topology and device capacity or model of the IES, that is, the planning of the IES starts from scratch. Secondly, based on the economical objective as well as efficient objective, an optimized configuration model for ECI was established. The advantage of this model is that it combines the energy dispatch optimization of the IES with the planning of the devices to support each other.
We have made major revision to the abstract.
Comment 3: The reviewer thinks it is important to compare with other methods proposed in the literature, highlighting the advantages and disadvantages of the proposed method.
Response 3: Thanks for the reviewer’s kindly suggestion. We have revised the whole manuscript carefully and tried to avoid any grammar or syntax error. In addition, we have asked several colleagues who are skilled authors of English language papers to check the English. We believe that the language is now acceptable for the review process.

Reviewer 3 Report
This paper presents an energy optimization for multi energy sources integration. The paper topic is valuable, however, to make this paper eligible for publications, it needs to be improved.
- The English language of this paper needs does not meet the publication standards. There are lots of grammatical errors and long sentences that are hard to follow.
- The problem statement is not clear. There are not sufficient related literature review that shows the need for this study.
- The optimization part is not clear. A flow chart is missing that show the details of the optimization.
- The cost between different energy sources and their conversions are not clear to the reader.
- The loads given as electrical, cold, heat are not clear. The cold and heat represents what type of load? How do we compare it with the electrical load? Figure 8 shows unit of MW on the y-axis for all the load types but the connection between the three loads are missing.
- It is very hard to follow Table 1 results.
- The conclusion is presented as the summary of the work presented. There is no information about what the take away of this study is. What do we learn from this study?
Some of the editorial comments:
- The abbreviations in Figure 7 are not given
- Section names are not appropriate.
- Circles in Fig 19 represents what?
- There is no attached table a mentioned in line 316
Author Response
Response to Reviewer 3 Comments
I am very grateful to your comments for the manuscript. According with your advice, we amended the relevant part in manuscript. We sincerely hope this manuscript will be finally acceptable to be published. The point to point responds to the comments are listed as following:
Comment 1: The English language of this paper needs does not meet the publication standards. There are lots of grammatical errors and long sentences that are hard to follow.
Response 1: Thanks for the reviewer’s kindly suggestion. We have revised the whole manuscript carefully and tried to avoid any grammar or syntax error. In addition, we have asked several colleagues who are skilled authors of English language papers to check the English. We believe that the language is now acceptable for the review process.
Comment 2: The problem statement is not clear. There are not sufficient related literatures review that shows the need for this study.
Response 2: We are very sorry that this part was not clear in the original manuscript.
At present, the optimization problems of regional integrated energy systems can be divided into two categories: 1) optimizing the capacity and model of the devices for a given regional IES; 2) optimizing the energy flow for existing system topologies. After reading a lot of literature, we found there are two difficulties in the study of IES: 1) simultaneous planning the topology and device capacity or model of the IES, that is, the planning of the IES starts from scratch; 2) combining the energy dispatch optimization of the IES with the planning of the devices to support each other.
Firstly, this paper establishes an energy input-output coupling model by deducing the energy input-output relationship of the energy conversion interface and classifying and refining the energy conversion devices. The model has good ductility and makes the relationship between energy flow and devices in the system become clear. Secondly, on the basis of economic indicators such as device installation cost, operation and maintenance cost, and annual cost, as well as energy-saving indicators such as energy utilization efficiency, an optimized configuration model for ECI is established. The model has the following innovations: 1) the planning of the IES starts from scratch; 2) energy optimization dispatch and device quantity are planned jointly.
We added this point into our revised manuscript.
Comment 3: The optimization part is not clear. A flow chart is missing that show the details of the optimization.
Response 3: Thanks for the reviewer’s valuable advice. Due to space limitations, we did not explain too much about the particle swarm algorithm in the original manuscript. But it is very necessary to give a flow chart of the optimization algorithm. We have added the flow chart in the manuscript (Figure A1).
Comment 4: The cost between different energy sources and their conversions are not clear to the reader.
Response 4: We are very sorry that this part was not clear in the original manuscript. We have added an appendix that gives the device-related parameters and energy prices for the reader's review.
Comment 5: The loads given as electrical, cold, heat are not clear. The cold and heat represent what type of load? How do we compare it with the electrical load? Figure 8 shows unit of MW on the y-axis for all the load types but the connection between the three loads are missing.
Response 5: We are very sorry that this part was not clear in the original manuscript. We classify the load according to the form of energy on the demand side. Thermal demands mainly include three aspects:space heating, hot water supplying and water-cycling system. Cooling demand includes space cooling. We have unified the units of the three load forms. We present the sankey diagram of the annual equipment output in the manuscript, as shown in Figure 11, from which we can see the coupling of various forms of energy.
Comment 6: It is very hard to follow Table 1 results.
Response 6: We are very sorry that the misunderstanding has been caused by the unclear expression of the original manuscript. Actually, the number planning of the device is equivalent to the capacity planning. The given capacity of the energy storage devices is so small, that the number of energy storage device is too large. If the capacity is increased, the number will decrease. Besides, in the future work, we will plan the location of the device in the energy conversion interface. Small-capacity devices are more conducive to it.
Comment 7: The conclusion is presented as the summary of the work presented. There is no information about what the take away of this study is. What do we learn from this study?
Response 7: We are very sorry that this part was not clear in the original manuscript.
Building an efficient integrated energy system will greatly improve the efficiency of energy system utilization and improve the economy of energy system. This paper focuses on two difficulties: 1) The initial planning of the IES which grows out of nothing; 2) Joint planning of energy dispatch optimization and device planning for IES.
The paper proposes an innovative energy conversion interface model, simplifies terminal IES into a multi-input to multi-output dual-port network. Firstly, according to the graph theory, the topological layering of system devices is carried out, mathematical relationship of energy input-output coupling is derived, and an overall model of the IES is obtained. Secondly, considering the dispatch optimization problem of IES, an optimized configuration model of energy conversation interface (ECI) is established with the goal of system economy and energy utilization efficiency. Finally, IES of a typical area is modeled and analyzed by the particle swarm optimization algorithm to optimize the configuration to improve the economy and energy utilization level. Installed capacity of devices under different optimization targets and device output level at each moment are obtained. Energy flow directions of the system devices are analyzed. Sensitivity analysis shows that system costs are most sensitive to energy prices, for example, natural gas price has the greatest impact on system economy in the study case. The weighted system optimization shows that as the economy proportion increases, the economy of the optimization result is better, and the energy utilization efficiency will be reduced. This provides a reference for decision making on IES planning.
We have revised the conclusions in the manuscript.
Comment 8: Some of the editorial comments:
1). The abbreviations in Figure 7 are not given.
2). Section names are not appropriate.
3). Circles in Fig 19 represents what?
4). There is no attached table a mentioned in line 316.
Response 8: Thanks for your careful review. We have modified the above issues in the manuscript.

Round 2
Reviewer 2 Report
The authors have addressed all my concerns regarding their research results and there is no further comment on the revised manuscript.
Reviewer 3 Report
The answers are satisfactory